# A Novel Method to Describe Large-Range Stress-Strain Relations of Elastic-Plastic Materials Based on Energy Equivalence Principle

**DOI:** 10.3390/ma16030892

**Published:** 2023-01-17

**Authors:** Simiao Yu, Lixun Cai, Ling Wang, Lin Lang

**Affiliations:** 1School of Architecture and Civil Engineering, Xihua University, Chengdu 610039, China; 2Applied Mechanics and Structure Safety Key Laboratory of Sichuan Province, School of Mechanics and Aerospace Engineering, Southwest Jiaotong University, Chengdu 610031, China

**Keywords:** uniaxial stress-strain relation, energy equivalence principle, elastic-plastic materials, large deformation analysis, semi-analytical model

## Abstract

Due to the unique structure of tensile sheet specimens with a circular hole (CHS specimen), a novel method is proposed to predict the large-range uniaxial stress-strain relations of elastic-plastic materials analytically. Based on the energy equivalence principle, a load-displacement semi-analytical model of the CHS specimen is proposed. Subsequently, a semi-analytical model of constitutive parameters of elastic-plastic materials is developed by virtue of the load-displacement relation of the CHS specimen, and the prediction of the material’s stress-strain relations is obtained. To examine the validity of the models, numerical simulations with a series of materials were performed. The results demonstrated that the dimensionless load-displacement curves and stress-strain relations obtained using the proposed models correspond well with those obtained using finite element analysis. In addition, tensile tests were performed on the CHS specimen for four elastic-plastic materials (T225 titanium alloy, 6061 aluminum alloy, Q345 steel, and 3Cr13 steel), and the validity of the models is also verified by the experimental results. Compared with the conventional uniaxial tensile tests, the stress-strain relation of elastic-plastic material captured by the novel method corresponds to a larger strain, which is of great importance for engineering design and safety assessment.

## 1. Introduction

The conventional tensile test is an important test to obtain the stress-strain relations of elastic-plastic materials, which can be used to characterize the elastic and plastic deformation degree of representative volume elements (RVEs). In uniaxial tensile tests, the load-displacement curves of the standard round bar (SRB) specimens are measured and converted to obtain the engineering stress-strain relations of the materials as
(1)$$\left\{ \begin{array}{l} \sigma_{E}=\frac{P}{A_{0}}\\ \varepsilon_{E}=\frac{L-L_{0}}{L_{0}} \end{array}\right.$$
where *σ*_E_ denotes the engineering stress, *ε*_E_ the engineering strain, *P* the load, *A*_0_ the initial cross-sectional area of the SRB specimens, *L*_0_ the initial gauge length, and *L* the gauge length after deformation. Considering the geometric deformation during the test, the engineering stress-strain curves can be converted into true stress-true strain curves as
(2)$$\left\{ \begin{array}{l} \sigma_{T}=\sigma_{E}(1+\varepsilon_{E})\\ \varepsilon_{T}=\ln(1+\varepsilon_{E}) \end{array}\right.$$
where *σ*_T_ denotes the true stress and *ε*_T_ the true strain. Equation (2) is determined on the basis of the assumption that the SRB specimen has uniform elongation deformation, but the RVEs of SRB specimens enter into a complex deformation state after necking which cannot satisfy this assumption. Thus, the accuracy of true stress-true strain curves after necking obtained from conventional tensile tests is questionable.

To produce the larger-range true stress-true strain curves of elastic-plastic materials, several researchers have presented mathematical methods. Bridgeman [1] assumed that the contour line of the necking section of the SRB specimen is an arc of radius *r*, and the strain on the cross-section of the necking section is a constant. Hence, the Bridgeman-modified formula is proposed as
(3)$$\left\{ \begin{array}{l} \varepsilon_{\mathrm{eq-Bridgeman}}=2\ln(a_{0}/a)\\ \sigma_{\mathrm{eq-Bridgeman}}=\frac{\sigma_{E}(1+\varepsilon_{\mathrm{eq-Bridgeman}})}{\left(1+2r/a)\ln(a/2r\right)} \end{array}\right.$$
where *a*_0_ denotes the initial radius of the gauge section, and *a* is the minimum radius of the gauge section during the deformation process. Some extended studies of the Bridgeman-modified formula have been conducted successively. Earl and Brown [2] and Earl [3] improved the precision of the Bridgeman-modified formula for specimens with small gaps. Through several tensile experiments on 18Ni steel, Chen [4] presented the assumption that the contour line of the necking section conforms to the hyperbola and proposed an analytical expression of the stress field on the smallest cross-section. In addition, the Bridgeman-modified formula is also applied to the studies on the necking behavior of sheet tensile specimens by Ling [5] and Hyun et al. [6]. Nevertheless, it can be proved that the assumptions of the Bridgeman-modified formula are unreasonable [7,8,9]; thus, the above studies are all inadequate to obtain the constitutive relations of elastic-plastic materials.

In recent decades, with the development of computer technology, computer-aided testing has been extensively used in experimental testing and data processing. Li [10] compared the load-displacement curves of uniaxial tensile tests with the finite element analysis (FEA) results and indicated the similarities and differences between them. Norris et al. [11] determined the average stress and average strain on the cross-section of the necking portion by FEA. P-Matic et al. [12,13,14] proposed a parameter iteration method to make the simulated load-displacement curves coincide with the experimental results, in which the strain-hardening exponent of material input into finite element software was adjusted continually. Zhang and Li [15,16] used the parameter iteration method for sheet specimens, and the obtained results are comparable to those of P-Matic. Built on the experiments and numerical analysis methods, Cabezas et al. [17] obtained the elastic moduli and hardening parameters of elastic-plastic materials, and the scope of application of these parameters under the plane stress state has been discussed. Michael [18] established a numerical model of the SRB specimens for large deformation analysis and compared the analysis results with the theoretical results. They conclude that the accuracy of the material constitutive relation obtained only by the calibration of a single specimen is questionable. Dumoulin et al. [19] studied the relation between the true stress-true strain curves and the load-displacement curves of the tensile specimens by observing the changes of the painted surface points and found that the strain growth in large deformation is not uniform. Mansoo et al. [20,21] proposed an iterative method with the experimental load-displacement curve as the target, and a non-defect necking simulation was realized based on the rigid plastic finite element theory, but this is difficult to implement using conventional finite element software. Based on the method of Mansoo et al., Kamaya and Wilkinson [22,23] presented a method to acquire the uniaxial constitutive relation of elastic-plastic materials in which the real-time true strain during the test was obtained through optical testing. However, considering the difference between the strain on the surface and the internal strain of the specimens, this method still needs to be further improved.

In 2010, Yao et al. [24] proposed the finite-element-analysis aided testing (FAT) method to obtain the full-range equivalent stress-strain relations of elastic-plastic materials. The FAT method is conducted based on the experimental load-displacement curves of notch round bar (NRB) specimens. The constitutive relations of the material entered into the finite element software are enhanced through a simple mathematical iteration in order to ensure that the experimental load-displacement curves of NRB specimens closely match the simulated results. Figure 1 shows the data processing flow chart for the FAT method. However, the FAT method must be implemented with the help of the FEA.

In addition, methods of finite element updating [25,26,27] and integrated digital image [28,29] have been widely used to obtain the constitutive relations of materials in recent years, but their main area of application is still linear elasticity.

The tensile sheet specimen with a circular hole (CHS specimen), which is shown in Figure 2, has a particular structure. The deformation of such specimens with a circular hole has always been a matter of concern. Siemen et al. [30] presented a method to analytically approximate the stress/strain field of an infinite plate under tension with a circular hole. Yao et al. [31] applied CHS specimens to the study of the critical fracture criterion for ductile materials. In the paper, a series of experiments and FEA were conducted to reveal the deformation characteristics of the CHS specimens. When the ratio of the radius *R* of the center hole to the width *W* ranges from 0.2 to 0.4; that is, 0.2 ≤ *R/W* ≤ 0.4, the RVE at the center point of the circular hole is under the uniaxial stress state during most of the deformation process, and only slight necking of the CHS specimen appears prior to fracture. The center point of the circular hole is also the crack initiation point of the CHS specimen. Figure 3 shows the deformation processes of three typical specimens obtained by FEA. The stress triaxiality *σ**, which is utilized to characterize the local stress state of the materials, of the RVE at the center point of the circular hole for CHS specimens is stable at around 1/3, and it corresponds to the uniaxial stress state of the materials. The stress triaxiality *σ** at the crack initiation point of the straight sheet specimen is gradually away from the uniaxial stress state after the large deformation occurred, while the stress triaxiality *σ** of the sheet specimen with a notch is always higher than 1/3 and gradually increases with the process of loading. Thus, it is applicable to use CHS specimens to characterize the uniaxial stress state of elastic-plastic materials.

In this paper, the energy equivalence method is employed to suggest a load-displacement semi-analytical model of CHS specimens, and a semi-analytical model of constitutive parameters of elastic-plastic materials is developed in virtue of the load-displacement relation of CHS specimens. Then, a novel method for obtaining the large-range true stress-true strain relations of elastic-plastic materials is presented. The precision of the method has been verified and is discussed in detail by experiments and FEA.

## 2. Theoretical Model

### 2.1. Energy Equivalence Principle

The most fundamental physical relation underlying continuum mechanics is the uniaxial stress-strain relation of the RVEs. The Ramberg–Osgood model [32] is frequently utilized to describe the elastic-plastic constitutive relations of elastic-plastic materials, which is expressed as
(4)$$\left\{ \begin{array}{l} \varepsilon_{\mathrm{eq}}=\varepsilon_{\mathrm{e-eq}}+\varepsilon_{\mathrm{p-eq}}\\ \varepsilon_{\mathrm{e-eq}}=\frac{\sigma_{\mathrm{eq}}}{E}\\ \varepsilon_{\mathrm{p-eq}}={\left(\frac{\sigma_{\mathrm{eq}}}{K}\right)}^{N} \end{array}\right.$$
where *ε*_eq_ denotes the equivalent total strain, *ε*_e-eq_ is the equivalent elastic strain, which is the ratio of the equivalent stress *σ*_eq_ to the elastic moduli *E*, *ε*_p-eq_ is the equivalent plastic strain, which can be expressed as a relationship between *σ*_eq_, the strain hardening exponent *N*, and the strength coefficient *K*. The Ramberg–Osgood model conforms to the principle of elastic-plastic strain additivity, which has been widely used by researchers [33,34,35,36,37,38].

The energy equivalence principle method was proposed by Chen and Cai [36,37,38] in 2017. On basis of the von Mises energy equivalence and integral mean value theorem, it is assumed that a loaded structure has a median point M where the average strain energy of the structure is equal to the strain energy density of the RVE at M. Also, the strain energy density of RVE at M under the complex stress state is equal to that under the equivalent uniaxial stress state. Further, the semi-analytical description of strain energy connected to geometry dimensions, constitutive relation parameters of materials, load, or displacement can be realized through derivation.

Based on the energy equivalence principle method [35,36,37], for a loaded structure of homogeneous and isotropic materials, its strain energy *U* is analytically expressed as
(5)$$\left\{ \begin{array}{l} U=\frac{V_{\mathrm{eff}}}{V^{*}}\int_{0}^{\varepsilon_{\mathrm{eq-M}}}f(\varepsilon_{\mathrm{eq}},E,K,N)d\varepsilon_{\mathrm{eq}}\\ \varepsilon_{\mathrm{eq-M}}=\phi \left(\frac{h}{h^{*}}\right)\\ \frac{V_{\mathrm{eff}}}{V^{*}}=\varphi \left(\frac{h}{h^{*}}\right) \end{array}\right.$$
where *ε*_eq_ denotes the equivalent strain, *ε*_eq-M_ is the equivalent strain of the RVE at M, *h* is the displacement, *h** is the characteristic displacement, *V*_eff_ is the effective deformation volume, and *V** is the characteristic volume. The geometric configuration of the loaded structure identifies the forms of *h** and *V**. Thus, the strain energy *U* can be represented analytically once the specific forms of the functions *f*, *ψ*, and *ϕ* are determined.

### 2.2. Linear Elastic Load-Displacement Model

On the linear elastic deformation stage of a loaded structure, the function *f* in Equation (5) can be expressed as
(6)$$\sigma_{\mathrm{eq}}=f(\varepsilon_{\mathrm{eq}},E,K,N)=E\varepsilon_{\mathrm{e-eq}}^{}$$

Thus, the elastic strain energy *U*_e_ can be obtained as
(7)$$U_{e}=\mu_{\mathrm{eM}}V_{\mathrm{e-eff}}=\frac{EV^{*}}{2}k_{1}{\left(\frac{h_{e}}{h^{*}}\right)}^{k_{2}}{\left[\phi \left(\frac{h_{e}}{h^{*}}\right)\right]}^{2}$$
where *h*_e_ denotes the elastic displacement, and *V*_e-eff_ is the elastic effective deformation volume.

Based on a series of calculations, the characteristic volume *V** of CHS specimens can be assumed as *V** = *RB*(2*W* − *πR*)^m^, where *m* is an effective volume reduction parameter, and the characteristic displacement *h** of CHS specimens can be assumed as *h** = *B*.

For a structure in which only elastic deformation occurs, the load (*P*) vs. displacement (*h*_e_) relation can be concluded as
(8)$$P=C_{e}h_{e}$$
where *C*_e_ denotes the elastic loading coefficient. The linear elastic energy *U*_e_ is integrated as part of the work-energy concept as
(9)$$U=W=\int_{0}^{he}Pdh=\frac{1}{2}C_{e}h_{e}{}^{2}$$

The linear elastic energy *U*_e_ of CHS specimens is calculated using the equality of Equations (7) and (9), and it is given as
(10)$$\left\{ \begin{array}{l} \frac{U_{e}}{U_{}^{*}}=\xi_{e}{\left(\frac{h_{e}}{h^{*}}\right)}^{2}\\ \xi_{e}=\frac{Ek_{0}}{2K}\\ U^{*}=KV^{*}=KRB{\left(2W-\pi R\right)}^{m}\\ h^{*}=B \end{array}\right.$$
where *ξ*_e_ denotes the curvature of dimensionless elastic energy vs. displacement curve, *U** is the characteristic energy, and *k*_0_ is the dimensionless elastic constant, which can be determined by a small amount of FEA.

Further, based on Castigliano’s theorem
(11)$$U=W=\int_{0}^{h}Pdh$$
the corresponding load-displacement relation of CHS specimens in elastic deformation is obtained as follows
(12)$$\left\{ \begin{array}{l} \frac{P}{P_{}^{*}}=2\xi_{e}(\frac{h_{e}}{h^{*}})\\ \xi_{e}=\frac{Ek_{0}}{2K}\\ P^{*}=KA^{*}=KB{\left(2W-\pi R\right)}^{m}\\ A^{*}=V^{*}/h^{*}=B{\left(2W-\pi R\right)}^{m}\\ h^{*}=B \end{array}\right.$$
where *P** denotes the characteristic load and *A** is the characteristic area.

### 2.3. Fully Plastic Load-Displacement Model

The function *f* in Equation (5) in fully plastic deformation can be described as
(13)$$\sigma_{\mathrm{eq}}=f(\varepsilon_{\mathrm{eq}},E,K,N)=K\varepsilon_{\mathrm{p-eq}}^{1/N}$$

The fully plastic strain energy *U*_p_ is integrated by substituting Equation (13) into Equation (5) as follows
(14)$$U_{p}=\frac{NKV^{*}}{N+1}k_{1}{\left(\frac{h_{p}}{h^{*}}\right)}^{k_{2}}{\left[\phi \left(\frac{h_{p}}{h^{*}}\right)\right]}^{1+1/N}$$
where *h*_p_ denotes the plastic displacement, *k*_1_ is the plastic effective volume coefficient, and *k*_2_ is the plastic effective volume exponent.

According to the research [37], the function *ϕ* satisfies power-law under fully plastic conditions as follows
(15)$$\phi \left(\frac{h}{h^{*}}\right)=k_{3}{\left(\frac{h_{p}}{h^{*}}\right)}^{k_{4}}$$
where *k*_3_ denotes the plastic effective strain coefficient and *k*_4_ is the plastic effective strain exponent.

Equation (15) is inserted into Equation (14) to produce the fully plastic energy *U*_p_ which is derived as
(16)$$\left\{ \begin{array}{l} \frac{U_{{}_{p}}^{}}{U_{{}_{}}^{*}}=\xi_{p}{\left(\frac{h_{p}}{h^{*}}\right)}^{m_{p}+1}\\ m_{p}=\frac{k_{4}}{N}+k_{4}+k_{2}-1\\ \;\xi_{p}=\frac{Nk_{1}k_{3}{}^{1/N+1}}{N+1}\\ U^{*}=KV^{*}=KRB{\left(2W-\pi R\right)}^{m}\\ h^{*}=B \end{array}\right.$$
where *m*_p_ denotes the plastic loading exponent and *ξ*_p_ is the curvature of the dimensionless fully plastic energy vs. displacement curve.

As with Equation (12), the load-displacement relation of CHS specimens under a fully plastic condition is obtained as
(17)$$\left\{ \begin{array}{l} \frac{P}{P_{}^{*}}=\left(1+m_{p}\right)\xi_{p}{\left(\frac{h_{p}}{h^{*}}\right)}^{m_{p}}\\ m_{p}=\frac{k_{4}}{N}+k_{4}+k_{2}-1\\ \;\xi_{p}=\frac{Nk_{1}k_{3}{}^{1/N+1}}{N+1}\\ P^{*}=KA^{*}=KB{\left(2W-\pi R\right)}^{m}\\ A^{*}=V^{*}/h^{*}=B{\left(2W-\pi R\right)}^{m}\\ h^{*}=B \end{array}\right.$$

### 2.4. Elastic-Plastic Load-Displacement Model

Generally, the real deformation of CHS specimens is elastic-plastic. Based on the engineering additivity principle, the displacement under elastic-plastic conditions can be expressed as
(18)$$h(P)=h_{e}(P)+h_{p}(P)$$

Substituting Equations (12) and (17) into Equation (18), the elastic-plastic load-displacement relation is obtained as
(19)$$\left\{ \begin{array}{l} \frac{h}{h^{*}}=\frac{1}{2\xi_{e}}\left(\frac{P}{P_{}^{*}}\right)+{\left(\frac{1}{\left(1+m_{p}\right)\xi_{p}}\left(\frac{P}{P_{}^{*}}\right)\right)}^{\frac{1}{m_{p}}}\\ \xi_{e}=\frac{Ek_{0}}{2K}\\ m_{p}=\frac{k_{4}}{N}+k_{4}+k_{2}-1\\ \;\xi_{p}=\frac{Nk_{1}k_{3}{}^{1/N+1}}{N+1}\\ P^{*}=KA^{*}=KB{\left(2W-\pi R\right)}^{m}\\ A^{*}=V^{*}/h^{*}=B{\left(2W-\pi R\right)}^{m}\\ h^{*}=B \end{array}\right.$$

Equation (19) is defined as the elastic-plastic load-displacement model for the CHS specimen (CHS-LD model), which can be utilized to analytically describe the load-displacement relations of the CHS specimens. In other words, if the material’s constitutive relationship is known, the load-displacement relation of the CHS specimen can be obtained on the basis of the CHS-LD model.

### 2.5. Acquisition of Uniaxial Stress-Strain Relation

If the load-displacement relations of the CHS specimens and the parameters of the CHS-LD model are known, the constitutive parameters *K*, *E*, and *N* can be achieved by inverse calculation.

First, the load-displacement relations of CHS specimens can be described as
(20)$$P=\left\{ \begin{array}{l} S_{e}\frac{h}{h^{*}}\;\;\;\;\;\;\;\;\;\;\;\;\;h\leq h_{y}\\ S_{p}{\left(\frac{h_{p}}{h^{*}}\right)}^{n_{p}}\;\;\;\;\;\;\;\;h>h_{y} \end{array}\right.$$
where *S*_e_ denotes the elastic stiffness, *S*_p_ is the plastic displacement coefficient, *n*_p_ is the plastic displacement exponent, and *h*_y_ is the yield displacement. *S*_e_, *S*_p_, and *n*_p_ are obtained through the regression of *P*-*h* curves, and the plastic displacement *h*_p_ is obtained from *h*_p_ = *h − h*/*S*_e_.

Substituting Equation (20) into Equation (19), the constitutive parameters of the materials can be represented as
(21)$$\left\{ \begin{array}{l} E=\frac{S_{e}}{A^{*}k_{0}}\\ N=\frac{k_{4}}{n_{p}-k_{4}-k_{2}+1}\\ K=\frac{S_{p}}{\left(1+m_{p}\right)\xi_{p}A^{*}} \end{array}\right.$$

Based on Equations (4) and (21), the uniaxial stress-strain relations of elastic-plastic materials can be obtained directly, and Equation (21) is referred to as the CHS-related stress-strain relation (CHS-SS) model. The maximum strain of the predicted uniaxial stress-strain relation corresponds to the maximum load point of the *P*-*h* curve.

Thus, a novel method is presented to obtain the large-range stress-strain relations of elastic-plastic materials. When the load-displacement relation of the CHS specimen is known, the constitutive parameters can be captured by means of the CHS-SS model, and the description of the large-range stress-strain relations of elastic-plastic materials is realized. This is a unique approach provided for the large-range stress-strain relations of elastic-plastic materials.

## 3. Verification of the Models

### 3.1. Parameters Determination by FEA

The commercial program ANSYS 14.5 was utilized to simulate the deformation process of the CHS specimens under three-dimensional conditions with different materials and geometric dimensions. The elements employed to model the CHS specimen are Solid 186, and the selected meshes are well-tested for convergence. Table 1 provides a summary of the details of the finite element models.

The parameters in the CHS-LD and CHS-SS models can be calibrated using a small amount of FEA. The displacement parameters used in the paper are those on the load line of the CHS specimens, that is, the displacement on the vertical center line as shown in Table 1. As shown in Figure 4, the effective volume reduction parameter *m* is calculated using the *P-h* curves of CHS specimens with different *R/W*, According to Equation (12), the parameter *k*_0_ can be established through a calculation under the elastic condition. Additionally, depending on Equation (17), the parameters *k*_1_–*k*_4_ are determined through at least two calculations under the plastic condition. The values of these parameters are given in Table 2.

### 3.2. Influence of R/W on the CHS-LD Model

A series of materials were adopted to simulate the elastic-plastic loading response of the CHS specimens with different *R/W* (i.e., *R/W* = 0.2, 0.25, 0.3, 0.35, 0.4) by FEA, where *E* was fixed at 210 GPa, *K* varied from 200 to 1600 MPa, whereas *N* varied from 4 to 10. As shown in Figure 5, it is apparent that the dimensionless load-displacement (*P/P**~*h/h**) curves are well concentrated together for each material, and these curves are all in excellent coincidence with the curves predicted by the CHS-LD model. This shows that the geometric parameter *R/W* has little influence on the CHS-LD model.

### 3.3. Effect of Constitutive Parameters on the CHS-LD Model

A series of numerical simulations were performed to find out how the constitutive parameters *K*, *E*, and *N* affected the CHS-LD model.

First, FEA simulations of CHS specimens of materials with various strain hardening exponents *N* were performed, where *E* was set at 210 GPa, *K* was set to 200, 400, 800, and 1600 MPa, and *N* was set to 4, 6, 8, 10, and 12. Taking the CHS specimens with *R/W* = 0.35 as an example, when the *E* and *K* are determined, it is illustrated that the trend of *P/P**~*h/h** curves with variable *N* is the same, but there are large differences in the value of the dimensionless loads, as shown in Figure 6. The dimensionless load *P/P** increases with *N* under the same condition of dimensionless displacement *h/h**.

The *P/P**~*h/h** curves of the materials with different strength coefficient *K* are shown in Figure 7, where *E* was fixed at 210 GPa, *N* was set to 4, 6, 8, and 10, and *K* was set to 200, 400, 800, and 1600 MPa. In addition, *R/W* = 0.35 was selected as the geometric dimension ratio of the CHS specimens. As shown in Figure 7, when the *E* and *N* are determined, the curves of the CHS specimens with different *K* deviate from each other, especially in the elastic stage. *K* has no effect on the linear elastic load-displacement curves of the CHS specimens, and the reason for this situation is that *K* is used in the dimensionless load *P/P** in Equation (12).

Figure 8 shows the *P/P**~*h/h** curves of the materials with different elastic moduli *E*, where *K* was set to 200, 400, 800, and 1600 MPa, *N* was set to 6 and 8, and *E* was set to 70, 110, and 210 GPa. Taking the CHS specimens with *R/W* = 0.35 as an example, it is illustrated that the *P/P**~*h/h** curves of the materials with the same *K* and *N* but variable *E* are different; however, they are very close to each other. The influence of *E* on the CHS-LD model is also concentrated on the elastic stage of the *P/P**~*h/h** curves.

In summary, all three constitutive parameters (*K*, *N*, *E*) have impacts on the CHS-LD model, in which *N* has the clearest effect, whereas the effect of *K* and *E* is mainly on the elastic stage of the *P/P*~h/h** curves. Furthermore, the above experiments all show that the *P/P*~h/h** curves predicted by the CHS-LD model correspond well with the simulated results of different materials. This proves that the CHS-LD model is stabilized and universal.

### 3.4. Verification of the CHS-SS Model

When the *P-h* curves of CHS specimens and the model parameters of the CHS-SS model are known, the analytical characterization of material constitutive parameters *K*, *N*, and *E* can be completed, and it is possible to fully determine the large-range stress-strain relations of elastic-plastic materials. The related stress-strain curves of materials are determined using the *P/P**~*h/h** curves in Figure 6. Comparing these stress-strain relations with the initial stress-strain relations input into the finite element software, as shown in Figure 9, two types of stress-strain curves are all in good correspondence with each other. Therefore, the sufficient accuracy and application of the CHS-SS model have been proved.

## 4. Experimental Verifications

### 4.1. Materials and Experimental Condition

To further verify the validity of the CHS-LD and CHS-SS models, tensile tests of CHS specimens fabricated from T225 titanium alloy, 6061 aluminum alloy, Q345 steel, and 3Cr13 steel were performed. The adopted materials all have stable mechanical properties and good ductility, and they are widely used in aviation, nuclear power, high-speed railway, and other important engineering projects. Table 3 lists the mechanical characteristics of the four types of materials together with the matching geometrical dimensions of the CHS examples.

The CHS specimens were subjected to tensile tests utilizing an MTS 809 servo-hydraulic testing system at room temperature and 0.01 mm/s displacement-control speed. To confirm the correctness and dependability of the results, several repeated experiments were carried out for each working condition on the specimens. To conserve space, the results of only two specimens are given for each working condition in the paper.

### 4.2. Verification of the CHS-LD Model

In this study, the ascent stage of the experimental *P-h* curves is analyzed. For the T225 titanium alloy, 6061 aluminum alloy, Q345 steel, and 3Cr13 steel, the experimental *P/P**~*h/h** curves of CHS specimens are shown in Figure 10. It can be seen that there is less dispersion of the *P/P**~*h/h** curves of each material. Furthermore, the *P/P**~*h/h** curves predicted by the CHS-LD model are all in good correspondence with the experimental *P/P**~*h/h** curves of the materials. The *P/P**~*h/h** curve of 6161 aluminum alloy obtained by the CHS-LD model displays a somewhat larger deviation from the experimental *P/P**~*h/h** curves at the initial plastic stage, but which is nevertheless smaller by less than 6%. For all four materials, the average absolute relative errors between the two types of *P/P**~*h/h** curves are less than 2%. Thus, the validity of the CHS-LD model is verified once more.

### 4.3. Verification of the Novel Method to Obtain Stress-Strain Relation

Based on the experimental *P/P*~h/h** curves in Figure 10, the uniaxial stress-strain relations of the T225 alloy, 6061 aluminum alloy, Q345 steel, and 3Cr13 steel were obtained using the novel method proposed in the paper. The predicted maximum strain is obtained from the maximum load point of the experimental *P-h* curve. Figure 11 compares the three types of stress-strain relations obtained from the conventional uniaxial tensile tests, the FAT method, and the novel method, respectively. The range of uniaxial stress-strain relations obtained using the FAT method is the largest, followed by those predicted by the novel model, whereas the range of the curves obtained from the conventional uniaxial tensile tests is the smallest. Moreover, the predicted uniaxial stress-strain relations of the T225 titanium alloy, Q345 steel, and 3Cr13 steel are in excellent correspondence with the stress-strain relations obtained from the conventional uniaxial tensile tests and the FAT method, and the average absolute relative errors between the predicted curves and the other two types of curves are less than 4%. Nevertheless, the forecast uniaxial stress-strain curve of the 6061 aluminum alloy deviates slightly from the other two curves when the material just enters the yield, and the maximum error is 12%. This happens because there is a yield platform in the constitutive relation of the 6061 aluminum alloy, which cannot be described accurately using the Ramberg–Osgood model, and it can be corrected by changing the form of the functions *f* in Equation (5).

The advantage of the novel method proposed in the paper over the FAT method is that the novel method realizes the theoretical characterization of constitutive relations of elastic-plastic materials, and only a few finite element calculations are needed to determine the parameters in the models. Moreover, the obtained large-range stress-strain relations are more accurate than the results given by extending the true stress-true strain curves of the conventional uniaxial tensile tests directly.

## 5. Conclusions and Discussions

In the study, CHS specimens, in which the stress triaxiality *σ** of the RVE at the center point of the circular hole is stable around 1/3, are utilized to obtain the large-range uniaxial stress-strain relations of elastic-plastic materials analytically. First, a semi-analytical model (CHS-LD) is suggested to represent the relation between the load and displacement of a CHS specimen based on the energy equivalence principle. Further, a semi-analytical model (CHS-SS) to represent the material’s constitutive properties is provided. The analysis and comparison results support the following conclusions:(1)The numerical simulations of CHS specimens fabricated from various materials were conducted, and the *P/P*~h/h** curves obtained by the CHS-LD model are consistent with the FEA results. In addition, the geometric parameter *R/W* has little influence on the CHS-LD model, and *K* and *E* have the main effect on the elastic deformation stage of the CHS-LD model, whereas *N* has the clearest effect on the CHS-LD model.(2)Corresponding to the simulated load-displacement relations, the stress-strain relations of the materials predicted by the CHS-SS model match the initial stress-strain relations fed into finite element software well.(3)The *P/P*~h/h** curves predicted by the CHS-LD model are all in good correspondence with the experimental *P/P*~h/h** curves of the materials including T225 titanium alloy, 6061 aluminum alloy, Q345 steel, and 3Cr13 steel. Moreover, the average absolute relative errors between the two types of *P/P*~h/h** curves are all less than 2%.(4)Compared with conventional uniaxial tensile tests and the FAT method, the novel method proposed in the study achieves the analytic description of uniaxial stress-strain relations, and it only requires a small amount of FEA. In addition, the range of the uniaxial true stress-strain relations obtained using the novel method has been significantly increased compared with that of the conventional uniaxial tensile tests.

Furthermore, the novel method could be conducted with other constitutive models rather than just the Ramberg–Osgood constitutive model. Also, the models presented in the paper are applicable to acquire the load-displacement and stress-strain relations of other configuration specimens, as long as the corresponding characteristic functions (*h**, *V**, *A**, and *P**) are changed accordingly.

## Figures and Tables

**Figure 1 materials-16-00892-f001:**
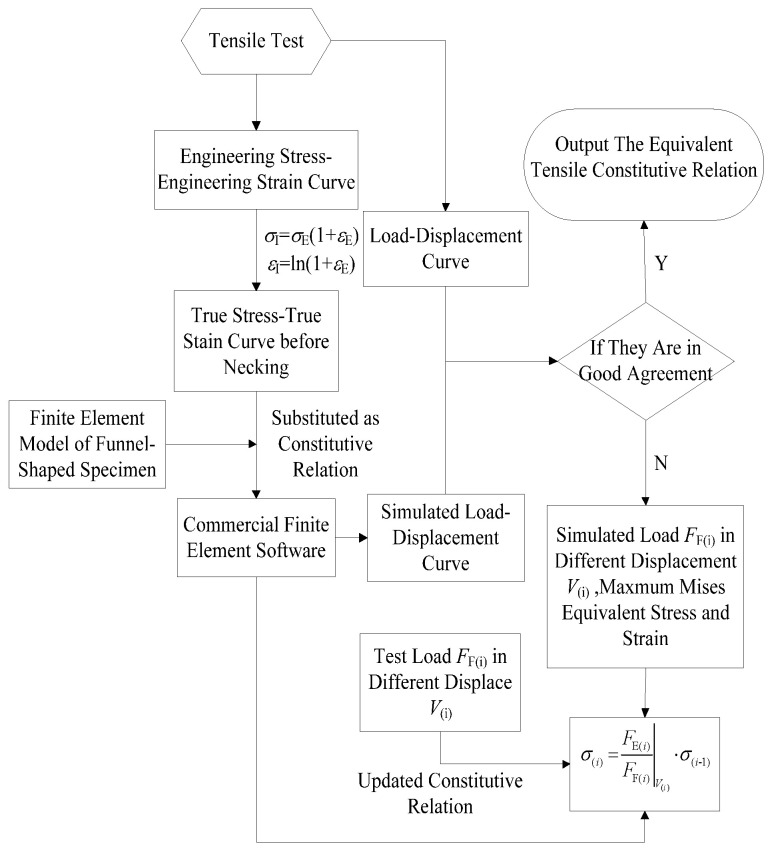
Flowchart for the FAT method’s data processing [24].

**Figure 2 materials-16-00892-f002:**
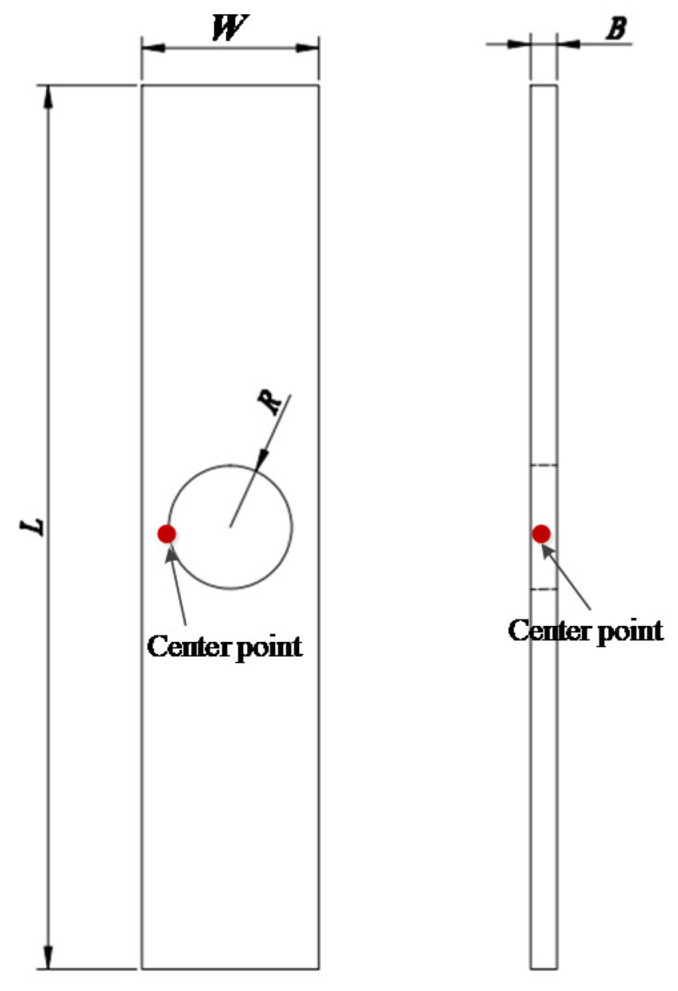
CHS specimen [31].

**Figure 3 materials-16-00892-f003:**
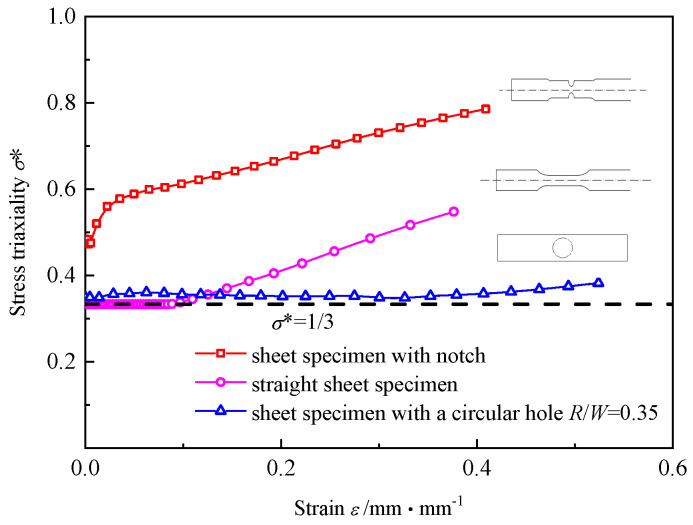
Evolution of stress triaxiality at the crack initiation point of different tensile specimens.

**Figure 4 materials-16-00892-f004:**
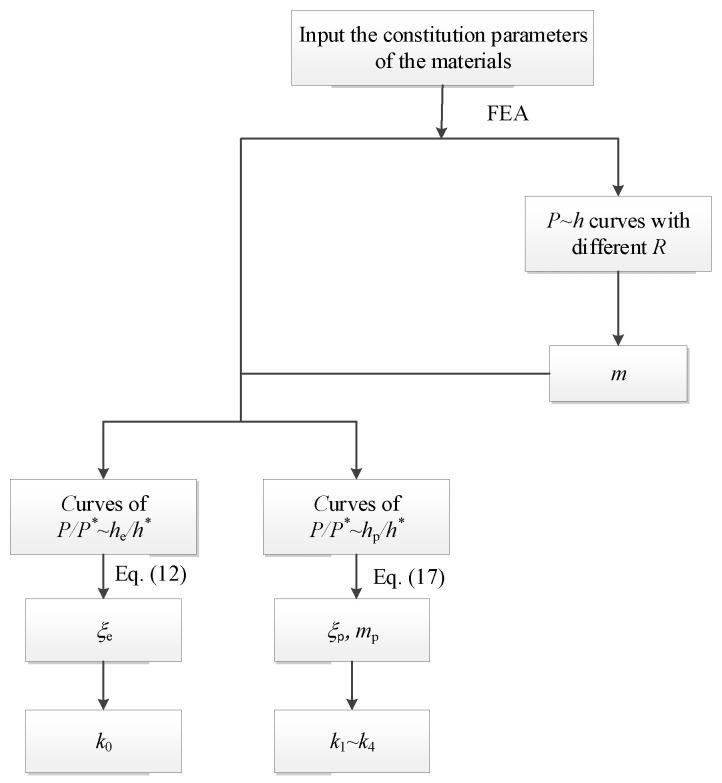
Procedure for the determination of the parameters of the models.

**Figure 5 materials-16-00892-f005:**
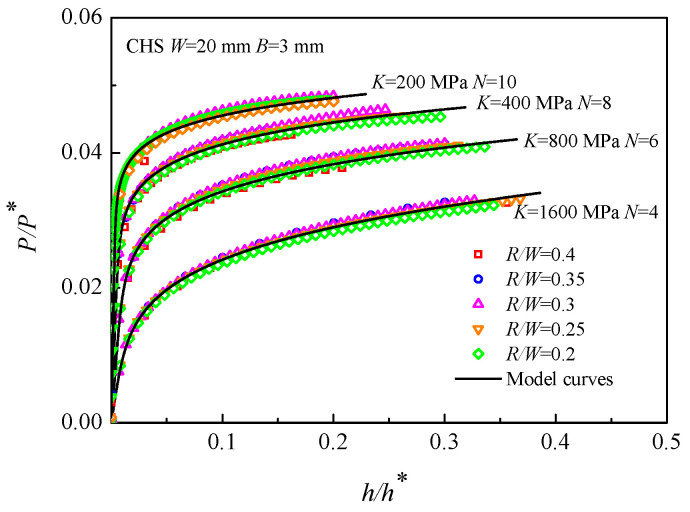
*P/P**~*h/h** curves of the CHS specimens with different *R/W*.

**Figure 6 materials-16-00892-f006:**
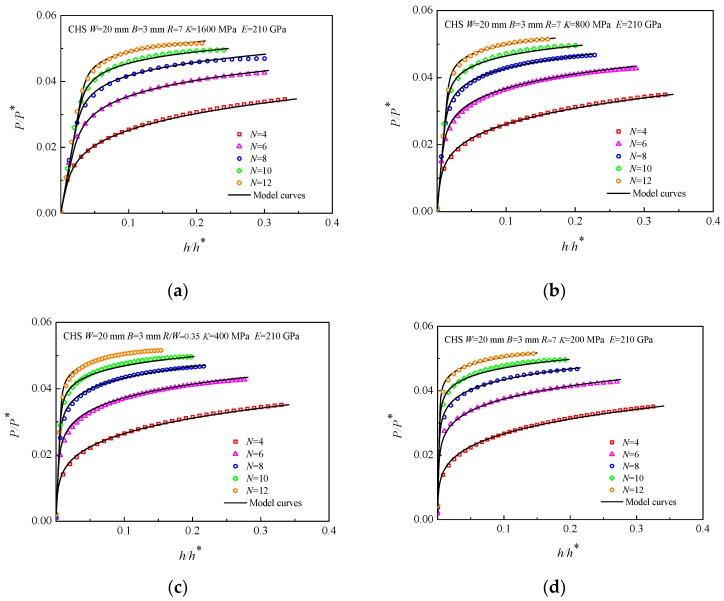
*P/P**~*h/h** curves of the CHS specimens with strain hardening exponent *N*, (**a**) *K* = 1600 MPa *E* = 210 GPa; (**b**) *K* = 800 MPa *E* = 210 GPa; (**c**) *K* = 400 MPa *E* = 210 GPa; (**d**) *K* = 200 MPa *E* = 210 GPa.

**Figure 7 materials-16-00892-f007:**
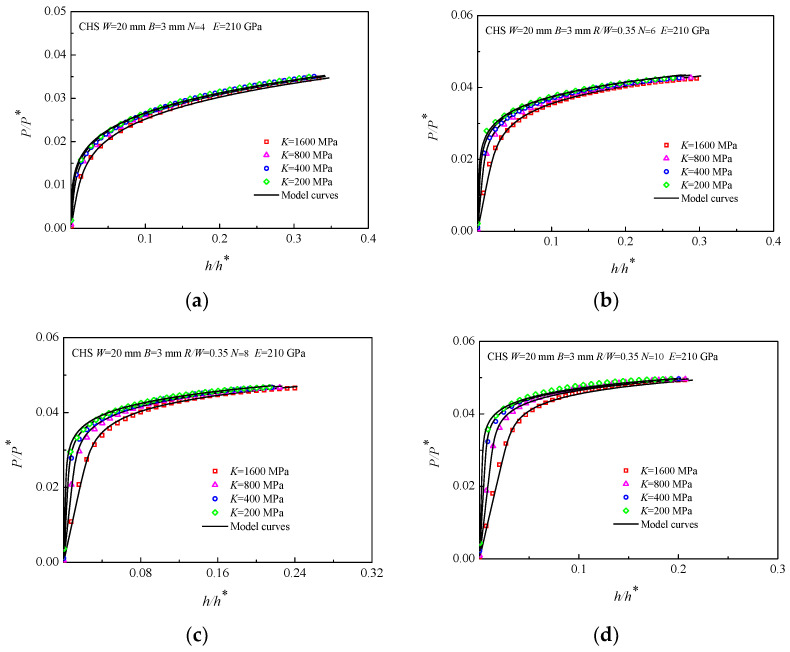
*P/P**~*h/h** curves of the CHS specimens with strength coefficient *K*, (**a**) *E =* 210 GPa *N =* 4; (**b**) *E =* 210 GPa *N =* 6; (**c**) *E =* 210 GPa *N =* 8; (**d**) *E =* 210 GPa *N =* 10.

**Figure 8 materials-16-00892-f008:**
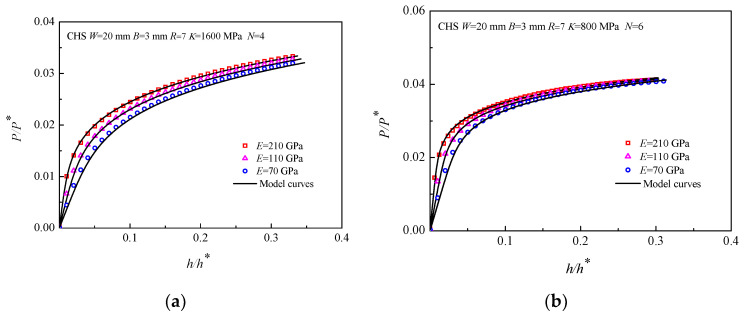
*P/P**~*h/h** curves of the CHS specimens with different elastic moduli *E,* (**a**) *K =* 1600 MPa *N =* 4; (**b**) *K =* 800 MPa *N =* 6; (**c**) *K =* 400 MPa *N =* 8; (**d**) *K =* 200 MPa *N =* 10.

**Figure 9 materials-16-00892-f009:**
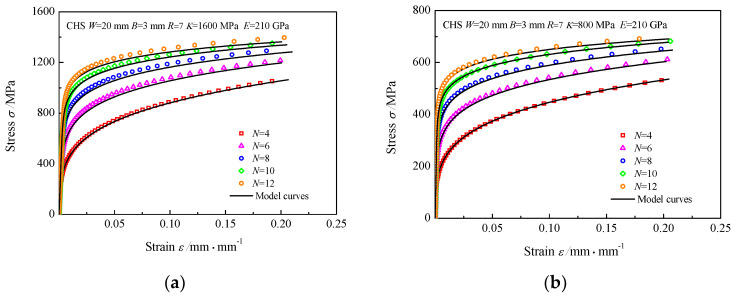
Predicted uniaxial stress-strain curves of materials, (**a**) *K* = 1600 MPa *E* = 210 GPa; (**b**) *K* = 800 MPa *E* = 210 GPa; (**c**) *K* = 400 MPa *E* = 210 GPa; (**d**) *K* = 200 MPa *E* = 210 GPa.

**Figure 10 materials-16-00892-f010:**
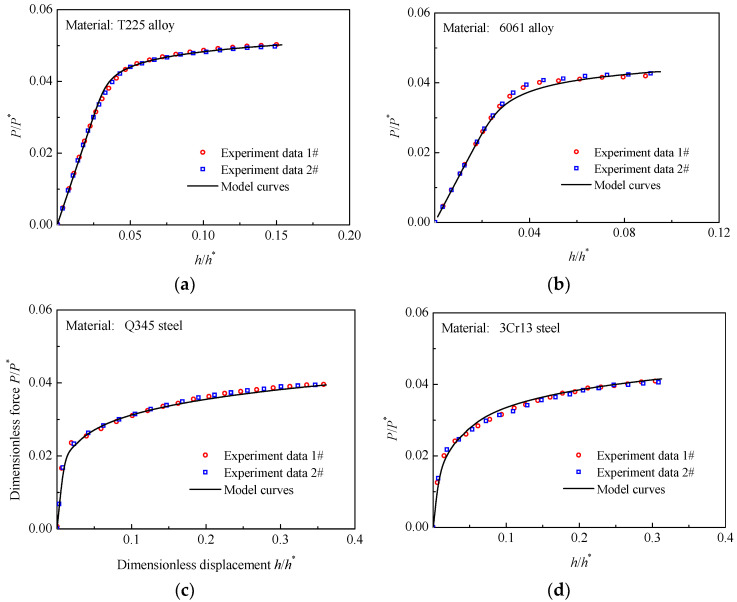
Experimental and Predicted *P/P*~h/h** curves of the CHS specimens; (**a**) T225 titanium alloy; (**b**) 6061 aluminum alloy; (**c**) Q345 steel; (**d**) 3Cr13 steel.

**Figure 11 materials-16-00892-f011:**
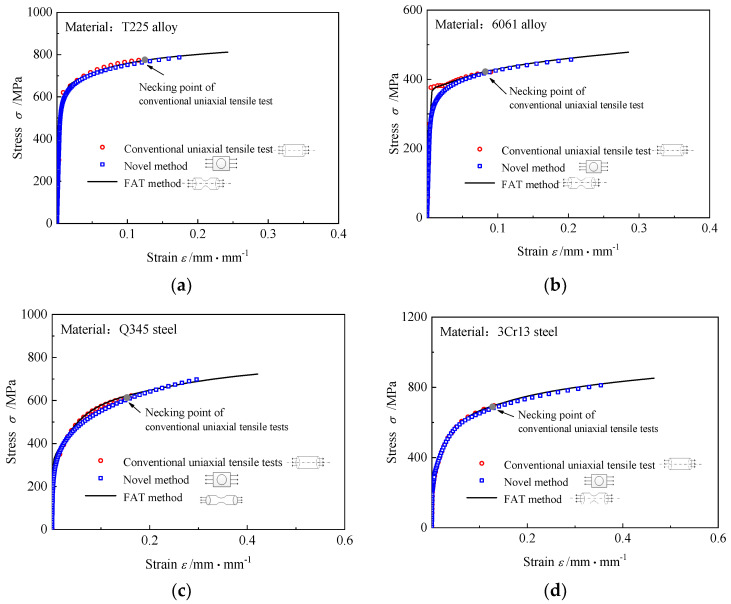
Comparison of the stress-strain curves obtained using three types of methods, (**a**) T225 titanium alloy; (**b**) 6061 aluminum alloy; (**c**) Q345 steel; (**d**) 3Cr13 steel.

**Table 1 materials-16-00892-t001:** Finite element models of the CHS specimen.

Element Model	Number of Nodes	Number of Elements	Configuration Size
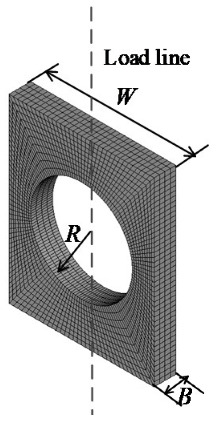	35,903 (*R* = 7 mm)	29,748 (*R* = 7 mm)	*W* = 20 mm *B =* 3 mm *R =* 4 mm, 5 mm, 6 mm, 7 mm, 8 mm

**Table 2 materials-16-00892-t002:** Value of the parameters of the models.

	*m*	*k* _0_	*k* _1_	*k* _2_	*k* _3_	*k* _4_
CHS	1.80	0.00901	0.450	0.00102	0.161	0.980

**Table 3 materials-16-00892-t003:** Mechanical parameters and corresponding geometric dimensions of the materials.

Materials	*E*/GPa	*K*/MPa	*N*	*W*/mm	*R*/mm	*B*/mm
T225	110	907	13.2	20	7.5	3
6061	72.6	453	14.3	20	7	3
Q345	209	829	6.45	20	7.5	3
3Cr13	204	969	6.12	20	7.5	3

## Data Availability

Not applicable.

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
