# Peer review of "A Novel Method to Describe Large-Range Stress-Strain Relations of Elastic-Plastic Materials Based on Energy Equivalence Principle"

_materials, 2023, doi:10.3390/ma16030892_

Round 1

Reviewer 1 Report

(1) lines 135, 205: "superposition" should be exchanged by "additivity" in this context, since mathematical homogenity F(ax)=aF(x) does not apply to the plastic strain

(2) lines 148, 158, 188: what is meant by "load line"?

(3) the Fig. in Tab. 1 shows much less than the reported 29748 elements 

(4) which material model was used in the ANSYS analyses? ANSYS 14.5 does not provide the Ramberg-Osgood model

(5) References: specifying the doi would be good

Reviewer 2 Report

The proposed method in this research to estimate the large range stress-strain relation is novel and applicable for tensile tests. Although the paper is well organized and the results are well described, some minor modifications are needed as follows:

1- the authors did not review the papers which used the CHS specimens for the tensile tests including this paper:

https://doi.org/10.1016/j.marstruc.2022.103205

2- The authors should mention the reference for Fig. 1.

3- It is clear that the CHS specimen has the lowest stress triaxiality and this reason made this sample a good candidate for this research, but what is the reference for Fig. 3?  Is this figure one of your calculated figures? If yes, the authors should mention this figure in the result section.

4- To evaluate the accuracy of the proposed model, authors could predict errors between the predicted flow stress and experimental flow stress values in Fig. 10. In this regard, standard statistical parameters such as correlation coefficient and average absolute relative error can be utilized. 

Reviewer 3 Report

Dear authors, you have nicely presented your article with experimental support. I actually enjoyed reading it.

My only concern is why experimental datapoints are not presented with error bars? This would allow to visually check on the efectiveness of the theoretical method.

Minor and less important:

I would change the word "realized" on line 16th by "obtained". Also an e'mail sould be added in line 9.

Reviewer 4 Report

The manuscript is discussing the large-range stress-strain relations of elastic-plastic materials based on energy equivalence principle. The results are interesting but I have some concerns summarized as following:

1) The introduction should have more related references. For instance, Figure 1, Fig.2 and Figure 3 titles should have the reference source.

2) From what I observe from Fig. 10, I believe the tensile tests performed are just 2 samples which I believe is not enough. Kindly, can the authors present in the experimental part the number of tensile tests. If it was just two please I recommend more tested samples as two only are not enough to represent the results. 

3) Conclusion should highlight the findings of the manuscript not repeating the results. please check the 3rd point of your conclusion.
